# Differential and Overlapping Effects of Melatonin and Its Metabolites on Keratinocyte Function: Bioinformatics and Metabolic Analyses

**DOI:** 10.3390/antiox10040618

**Published:** 2021-04-17

**Authors:** Joanna Stefan, Tae-Kang Kim, Fiona Schedel, Zorica Janjetovic, David K. Crossman, Kerstin Steinbrink, Radomir M. Slominski, Jaroslaw Zmijewski, Meri K. Tulic, Russel J. Reiter, Konrad Kleszczyński, Andrzej T. Slominski

**Affiliations:** 1Department of Dermatology, University of Alabama at Birmingham, Birmingham, AL 35294, USA; jstefan109@gmail.com (J.S.); tkkim4567@gmail.com (T.-K.K.); zjanjetovic@uabmc.edu (Z.J.); radomir.slominski@gmail.com (R.M.S.); 2Department of Oncology, Nicolaus Copernicus University Medical College, Romanowskiej Str. 2, 85-796 Bydgoszcz, Poland; 3Department of Dermatology, University of Münster, Von-Esmarch-Str. 58, 48149 Münster, Germany; Fiona.Schedel@ukmuenster.de (F.S.); kerstin.steinbrink@ukmuenster.de (K.S.); konrad.kleszczynski@ukmuenster.de (K.K.); 4Department of Genetics, Comprehensive Cancer Center, University of Alabama at Birmingham, Birmingham, AL 35294, USA; dcrossman@uabmc.edu; 5Department of Medicine, Comprehensive Cancer Center, University of Alabama at Birmingham, Birmingham, AL 35294, USA; jaroslawzmijewski@uabmc.edu; 6Team 12, Centre Méditerranéen de Médecine Moléculaire (C3M), Université Côte d’Azur, INSERM U1065, 06200 Nice, France; meri.tulic@unice.fr; 7Department of Cellular and Structural Biology, UT Health Science Center, San Antonio, TX 78229, USA; reiter@uthscsa.edu; 8Pathology and Laboratory Medicine Service, VA Medical Center, Birmingham, AL 35294, USA

**Keywords:** melatonin, metabolites of melatonin, RNA-*sequencing*, human keratinocytes, mitochondrial metabolism

## Abstract

We investigated the effects of melatonin and its selected metabolites, i.e., *N*^1^-Acetyl-*N*^2^-formyl-5-methoxykynurenamine (AFMK) and 6-hydroxymelatonin (6(OH)Mel), on cultured human epidermal keratinocytes (HEKs) to assess their homeostatic activities with potential therapeutic implications. RNA*seq* analysis revealed a significant number of genes with distinct and overlapping patterns, resulting in common regulation of top diseases and disorders. Gene Set Enrichment Analysis (GSEA), Reactome FIViZ, and Ingenuity Pathway Analysis (IPA) showed overrepresentation of the p53-dependent G1 DNA damage response gene set, activation of p53 signaling, and NRF2-mediated antioxidative pathways. Additionally, GSEA exhibited an overrepresentation of circadian clock and antiaging signaling gene sets by melatonin derivatives and upregulation of extension of telomere signaling in HEKs, which was subsequently confirmed by increased telomerase activity in keratinocytes, indicating possible antiaging properties of metabolites of melatonin. Furthermore, Gene Ontology (GO) showed the activation of a keratinocyte differentiation program by melatonin, and GSEA indicated antitumor and antilipidemic potential of melatonin and its metabolites. IPA also indicated the role of Protein Kinase R (PKR) in interferon induction and antiviral response. In addition, the test compounds decreased lactate dehydrogenase A (*LDHA*) and lactate dehydrogenase C (*LDHC*) gene expression. These results were validated by qPCR and by Seahorse metabolic assay with significantly decreased glycolysis and lactate production under influence of AFMK or 6(OH)Mel in cells with a low oxygen consumption rate. In summary, melatonin and its metabolites affect keratinocytes’ functions via signaling pathways that overlap for each tested molecule with some distinctions.

## 1. Introduction

Melatonin, a tryptophan (Trp) metabolism product, is widely detected in many species, including animals, plants, unicellular eukaryotes, algae, bacteria and humans [1,2]. It regulates the circadian rhythm and exerts pleiotropic bioactivities mediated by interactions with high affinity G-protein coupled membrane bound melatonin receptors type 1 and type 2 (MT1 and MT2) or receptor-independent mechanisms [3,4]. Melatonin also acts as a broad-spectrum antioxidant, stimulator of antioxidative responses, and of DNA repair pathways at relatively high concentrations (>1 µM) [5,6]. It also has anticancer activities [7,8,9,10], which are shared by products of its metabolism [11,12,13,14,15,16].

Melatonin, mainly produced in the pineal gland, is also synthesized in peripheral organs, such as the skin [13,17,18,19,20,21,22]. This led to our proposal that the cutaneous melatoninergic system can defend this organ against environmental damage, with mitochondria playing an important role in this function [10,11,21,22,23], while being a part of a complex cutaneous neuroendocrine system that coordinates cutaneous responses to stress [4,24], including ultraviolet radiation (UVR) [23]. Extensive data from various laboratories [10,11,12,16,17,22,23,25,26,27,28] have documented that melatonin and/or its metabolites induce potent antioxidative actions, using DNA repair mechanisms against UVB-induced damages in human cutaneous cells including stimulation of NRF2 and p53-dependent pathways that are independent from MT1 and MT2 [12,16,28]. Importantly, melatonin and its metabolites accumulate in the human epidermis [29,30] and melatonin exerts anti-erythema and antiaging effects when applied topically to the skin [31,32].

Melatonin is also recognized for its complex immunomodulatory [33,34,35] and antiviral activities [36,37,38]. Specifically, it can regulate T-cell and macrophage functions, and possesses both immunostimulatory and anti-inflammatory properties that are context-dependent [33,35,39]. Thus, melatonin is a double-edged sword, being in some cases beneficial, while in others aggravating autoimmune responses [33,35,39,40,41]. Its anti-inflammatory action includes inhibition of NO synthases, TLR4 and NF-κB activation, as well as the upregulation of NRF2 [39]. Many of melatonin’s anti-inflammatory effects appear to be mediated by sirtuin-1 (SIRT1) [39,40]. Lastly, melatonin itself and its metabolites affect mitochondrial functions and energy yielding cellular metabolism, including in skin cells [1,9,14,16,22,42]. In addition, antiaging skin properties of melatonin are widely recognized [9,31,39,43].

To provide insight into the mechanisms of the above antioxidative, protective, metabolic, prodifferentiation, antiaging, anti-inflammatory, and antitumor mechanisms in the epidermis, we conducted RNA*seq* analyses using human epidermal keratinocytes treated with melatonin and its indolic (6-hydroxymelatonin) and kynuric *N*^1^-Acetyl-*N*^2^-formyl-5-methoxykynurenamine (AFMK) metabolites. We found overlapping and differential effects that were further investigated using biochemical and molecular assays. By using the comprehensive Omics method of estimation of gene expression, we obtained an insight not only into the expression of particular genes involved in the above processes, but also the correlation between these genes and their functional clustering. Based on these analyses, the ability of melatonin’s metabolites to regulate different pathways affecting the physiological and pathological states of epidermis remains under discussion.

## 2. Materials and Methods

### 2.1. Reagents

2-Hydroxymelatonin (2(OH)Mel), 6-hydroxymelatonin (6(OH)Mel), bovine serum albumin (BSA), ethanol (EtOH), glucose, melatonin, and sodium pyruvate were purchased from Sigma (St. Louis, MO, USA). AFMK and GlutaMAX™ supplements were purchased from Cayman Chemical (Ann Arbor, MI, USA) and Thermo Fisher Scientific (Waltham, MA, USA), respectively. Other reagents were supplied as follows: EpiGRO™ Human Epidermal Keratinocyte Complete Culture Media Kit (Millipore Merck KGaA, Darmstadt, Germany); RNA Miniprep Kit (Agilent Technologies, Santa Clara, CA, USA), high capacity cDNA Reverse Transcription Kit (Applied Biosystems), DyNamo Flash SYBR Green qPCR Kit (Thermo Scientific, Waltham, MA, USA), KAPA SYBR^®^ FAST qPCR Kit (Kapa Biosystems, Wilmington, MA, USA), the Cell Mito Stress Test media was supplemented with 2 mM GlutaMAX™ Supplement (*L*-alanyl-*L*-glutamine dipeptide in NaCl), Proteome Profiler Array kit (R&D Systems, Minneapolis, MN, USA), lactate colorimetric assay kit (Cell Biolabs, Inc. San Diego, CA, USA), TeloTAGGG Telomerase Elisa assay (Roche, Basel, Switzerland), and a set of human primers for real-time PCR (Eurofins Genomics, Ebersberg, Germany).

### 2.2. Cell Culture and Treatment

Neonatal Human Epidermal Keratinocytes (HEKn, primary cells) were isolated from foreskins collected at the Woman and Infant Hospital (UAB) and cultured on Petri dishes (TPP) in EpiGRO™ Human Epidermal Keratinocyte Complete Culture Media Kit (Millipore Merck KGaA, Darmstadt, Germany) as described previously [12]. Cells were seeded into 24-well plates, maintained until they reached 70% confluency, and stimulated for 24 h with melatonin, 6(OH)Mel or AFMK at the final concentration of 10^−5^ M or with solvent only (control, 0.1% EtOH). HaCaT cells were grown in Dulbecco’s Modified Eagle Medium (DMEM) media containing 5% charcoal stripped FBS followed by treatment with 10^−4^ or 10^−5^ M melatonin and 10^−5^ M 6(OH)Mel and 2-hydroxymelatonin (2(OH)Mel) at 37 °C under 5% CO_2_ for 6 and 24 h. We have used similar doses of ligands in our previous studies, which demonstrated the paracrine mechanism of action secondary to production and metabolism of melatonin in the epidermis [12,13,29,30]. After 6 and 24 h, cells were collected in lysis reagent and RNA isolated using Absolutely RNA Miniprep Kit (Agilent Technologies, Santa Clara, CA, USA).

### 2.3. Reverse Transcription Reaction and Real-Time PCR

To study gene expression in material acquired from HaCaT cells, cDNA was synthesized using 0.4 µg RNA with High Capacity cDNA Reverse Transcription Kit (Applied Biosystems, Foster City, CA, USA) following manufacturers’ protocols. KAPA SYBR^®^ FAST qPCR Kit (Kapa Biosystems, Wilmington, MA, USA) were used for qRT-PCR with Quant Studio 6 Real-Time PCR Systems (Applied Biosystems, Foster City, CA, USA) in presence of the following primers’ sequences: lactate dehydrogenase A (LDHA) (Fwd: 5′-ACCCAGTTTCCACCATGATT, Rev: 5′-CCCAAAATGCAAGGAACACT), PGC-1 (Fwd: 5′-GTCACCACCCAAATCCTTAT, Rev: 5’-ATCTACTGCCTGGAGACCTT), SIRT-1 (Fwd: 5′-TCGCAACTATACCCAGAACATAGACA, Rev: 5′-CTGTTGCAAAGGAACCATGACA), and cyclophilin B (Fwd: 5′-TGTGGTGTTTGGCAAAGTTC, Rev: 5′-GTTTATCCCGGCTGTCTGTC) as the house-keeping gene. The following temperature profile was set using Quant Studio 6 real-time PCR Systems (Applied Biosystems): hold stage (predenaturation): 95 °C for 20 s, PCR stage (40 cycles): (denaturation): 95 °C for 1 s, (annealing): 60 °C for 20 s.

### 2.4. RNA Sequencing and Bioinformatics Analysis

At least 200 ng of RNA from each HEKn sample with purity of 2.0 (A_260/280_) was sent for RNA sequencing by Novogene (Sacramento, CA, USA). Prior to sending, the sample quality and concentration were determined using Cytation 5 Imaging Reader. After sequencing, the raw sequence FASTQ files were trimmed to remove primer adapters using Trim Galore version 0.6.2 (parameters used: --paired; --trim1). The trimmed FASTQ files were then aligned to the Gencode GRCh38 p7 Release 25 genome using STAR version 2.7.1a (parameters used: --outReadsUnmappedFastx; --outSAMtype BAM SortedByCoordinate; --outSAMattributes All). Transcript abundances were then calculated from the alignments using HTSeq count version 0.11.2 (parameters used: -m union; -r pos; -t exon; -igene_id; -a 10; -s no; -f bam). The raw counts were then loaded into DESeq2 using their default settings to normalize and perform pairwise differential expression. All raw data were deposited in NCBI GEO (GSE147588).

### 2.5. Bioinformatics

Pathway analysis was performed using Reactome and Ingenuity Pathway Analysis software (Ingenuity^®^ Systems). Gene set enrichment analysis (GSEA) and diagram analysis with overlapping gene scores into pathways were made using Reactome FIViZ software. Data analyzed using IPA software were processed with the following steps: for generating networks, a data set containing gene identifiers and corresponding expression values were uploaded into the application. Each identifier was mapped to its corresponding object in Ingenuity’s Knowledge Base. A fold change cutoff of +/−2 was set to identify molecules whose expressions were significantly differentially regulated. These molecules, called Network Eligible molecules, were overlaid onto a global molecular network developed from information contained in Ingenuity’s Knowledge Base. Networks of Network Eligible Molecules were then algorithmically generated based on their connectivity. The functional analysis identified the biological functions and/or diseases that were most significant to the entire data set. Molecules from the data set that met the fold change cutoff of +/−2 and were associated with biological functions and/or diseases in Ingenuity’s Knowledge Base were considered for the analysis. Right-tailed Fisher’s exact test was used to calculate the *p* value determining the probability that each biological function and/or disease assigned to that data set is due to chance alone.

### 2.6. Analysis of Metabolic Function

Primary human keratinocytes were isolated from skin biopsies of healthy controls as previously described [44]. Keratinocytes were seeded at 0.25 × 10^5^ cells/well overnight on Seahorse 96-well XFe96 microplates in culture medium. The next day, cells were treated with melatonin, AFMK and 6(OH)Mel for 24 h, all at a final concentration of 10^−5^ M. Just before the start of the experiment, cells were depleted of glucose for 1 h in a 37 °C non-CO_2_ incubator. Oxygen consumption rate (OCR or mitochondrial respiration) and extracellular acidification rate (ECAR or glycolytic function) in live cells in real-time were measured using the XF Cell Mito Stress Test and Agilent Seahorse XF96 Extracellular Flux Analyzer (Seahorse Bioscience, North Billerica, MA, USA). The Cell Mito Stress Test media was supplemented with 2 mM GlutaMAX™ Supplement, 1 mM sodium pyruvate, and 25 mM glucose and injected according to the Mito Stress Test. Results are automatically generated from *Wave* data that was exported to Excel.

### 2.7. Proteome Profiler and L-Lactate Functional Assays

To measure cytokine release, conditioned media from HEKn cells collected after treatment with melatonin, AFMK or 6(OH)Mel for 24 h were submitted for human cytokine array assay using Proteome Profiler Array kit (R&D Systems). To evaluate *L*-lactate production in HEKn cell media, the Lactate Assay kit (Colorimetric, Cell Biolabs) was used and the values were recorded using Cytation 5 Imaging Reader. Telomerase activity was measured using the TeloTAGGG Telomerase Elisa assay (Roche) in the cell pellets obtained after 24 h treatment of HaCaT keratinocytes with AFMK, melatonin or 6(OH)Mel at a concentration of 10^−4^ or 10^−5^ M. The data were collected with Cytation 5 Imaging Reader. All assays followed procedures provided by manufacturers.

### 2.8. Statistical Analysis

Experiments were performed at least three times, with results expressed in each case as the mean + standard deviation (SD). Significant differences between results were determined by both by Mann–Whitney U test and Student *t*-test, and an appropriate post hoc analysis using GraphPad Prism 7.05 software (La Jolla, CA, USA). *p* value of less than 0.05 was considered statistically significant.

## 3. Results and Discussion

### 3.1. RNAseq Results

The analysis of RNA*seq* data with a fold change cutoff of +/−2 showed changes in the expressions of 6741 genes, 6831 genes and 6433 genes after respective treatments of keratinocytes with 6(OH)Mel, AFMK and melatonin. 6(OH)Mel, AFMK and melatonin enhanced expressions of 3735, 3721 and 3200 genes, and inhibited 3006, 3110 and 3233 genes, respectively (Figure 1A–D, Appendix A). An overlapping gene expression pattern included upregulation of 948 and downregulation of 957 common genes for all tested compounds. The Venn diagram also shows a distinct pattern of genes specific for each compound tested with 933 (14.4%), 1134 (17.5%) and 1155 (17.9%) genes upregulated in keratinocytes by melatonin, 6(OH)Mel and AFMK, respectively. Similarly, numbers of genes downregulated only by melatonin, 6(OH)Mel and AFMK, respectively, were 1050 (18.6%), 881 (15.6%) and 969 (17.2%).

### 3.2. Bioinformatics Analysis

IPA demonstrated common regulation of top three diseases and disorders including cancer, dermatological diseases and conditions and organismal injuries and abnormalities by all compounds, with two additional ones having different sequence orders: reproductive system disease and endocrine system disorders for melatonin, reproductive system disease and psychological disorders for 6(OH)Mel, and endocrine system disorders and gastrointestinal diseases for AFMK (Table 1, Appendix A). There were also similarities with some differences in regulation of molecular and cellular functions that included cell-to-cell signaling and interactions, gene expression, cell signaling, molecular transport and vitamin and mineral metabolisms for melatonin; molecular transport, cell-to-cell signaling and interaction, cell signaling, vitamin and mineral metabolisms and cell morphology for 6(OH)Mel; molecular transport, cellular function and maintenance, cell signaling, vitamin and mineral metabolisms and cellular movement for AFMK (Table 1).

GSEA and Reactome FIViZ revealed upregulation of gene sets connected with responses to oxidative stress and DNA repair (Table 2). IPA was consistent with this analysis by showing stimulation of the p53 pathway and p38 MAPK signaling, NER pathway, NRF2-mediated oxidative stress response by melatonin, AFMK and 6(OH)Mel and the role of BRCA1 in DNA damage response only by melatonin and AFMK (Figure 2; Figure 3).

The profile and Z-scores of expressions of the genes involved in NRF2 downstream signaling affected by tested compounds are shown in Figure 3A. Expression of solute carrier family 6 member 9 (*SLC6A9*) was stimulated by melatonin treatment, inhibited by 6(OH)Mel and did not change after AFMK treatment. Expression of early growth response protein 1 (*EGR1*) was upregulated by melatonin, downregulated by 6(OH)Mel and showed no change for AFMK, carboxylesterase 2-phase 1 protein (*CES2*), EPH receptor A2 (*EPHA2,* which belongs to ephrin receptor subfamily), biliverdin reductase B (*BLVRB*), and transforming growth factor α (*TGFA*) and had similar expression levels to the control after treatment with AFMK or melatonin and was downregulated by 6(OH)Mel. *SQSTM1* (sequestosome 1) had a similar Z-score for AFMK and 6(OH)Mel and the control and a negative Z-score value (-1.5) for melatonin. Both heparin binding EGF-like growth factor (*HBEGF*) and heme oxygenase 1 (*HMOX1*) genes were inhibited by the tested compounds. Expression of solute carrier family 7 member 11 (*SLC7A11*) and solute carrier family 2 member 3 (*SLC2A3*) was inhibited by melatonin and 6(OH)Mel, but not by AFMK. Thioredoxin reductase 1 (*TXNRD*1) was strongly upregulated by AFMK and to a lower degree by 6(OH)Mel and slightly by melatonin. This observation is consistent with reported role of AFMK in protection against oxidative stress [10,26,44]. Expression of neuregulin 1 (*NRG1*) was only stimulated by AFMK. Transporter solute carrier family 39 member 7 (*SLC39A7*) only had a positive Z-score for AFMK, and solute carrier family 39 member 4 (*SLC39A4*) only had a positive Z-score for 6(OH)Mel and melatonin. MAF bZIP transcription factor F (*MAFF*) and v-maf avian musculoaponeurotic fibrosarcoma oncogene homolog G (*MAFG*), crucial regulators of mammalian gene expression, were strongly upregulated only by melatonin with weak stimulation by AFMK and 6(OH)Mel for *MAFG* only. In summary, these analyses are consistent with previous experimental data showing that melatonin and its metabolites differentially regulate the Nrf2 signaling pathway in keratinocytes [12,23,27,36] and such regulation would depend on melatonin metabolism [22].

The profile and Z-scores of expressions of the genes involved in TP53 signaling are presented in Figure 3B. *TP53BP*1 and *TP53TG1* were upregulated by AFMK. The expression levels of growth arrest and DNA damage-inducible protein gamma (*GADD45G*) and *GADD45A* increased after melatonin treatment, and *TP53I3* and *TP53INP2* were upregulated by AFMK and 6(OH)Mel. The increased *GADD45G* level is consistent with melatonin involvement in cellular stress responses and DNA repair processes, as well as tumor suppression [1,10,12,21,26,27].

Increased *GADD45A* expression by melatonin was associated with DNA damage-induced transcription of this gene mediated by both p53-dependent and p53-independent mechanisms and responses to environmental stress by mediating activation of the p38/JNK pathway via MTK1/MEKK4 kinase. *P53I11* and *TP53INP1* were also upregulated by AFMK and 6(OH)Mel. All compounds increased the expression of *GADD45B,* indicating their involvement in the response to environmental stress by activation of p38/JNK signaling [45].

The expression profile of genes involved in a mitochondrial metabolism was similar for all tested compounds (Table 3). However, minor differences were seen in the regulation of pathway of glycogen synthesis, which was upregulated by melatonin and AFMK, but not by 6(OH)Mel. The mitochondrial fatty acid beta-oxidation signaling was over-represented only after melatonin treatment (Table 3). Selective investigation of genes from the tested pathways showed a decrease in *LDHA* expression after 6(OH)Mel (fold change: −1.02) AFMK (fold change: −1.03) melatonin (fold change: −1.05) treatments. Similarly, a decrease in lactate dehydrogenase C (*LDHC*) gene expression was seen (fold change: 6(OH)Mel: −3.22, AFMK: −2.07, melatonin: −1.19).

Therapeutic implications for melatonin and its metabolites based on the GSEA are shown in Table 4. GSEA shows an increase in extension of telomeres, telomere maintenance, telomere C-strand (lagging strand) synthesis and packaging of telomere end signaling, which indicates antiaging properties of melatonin, AFMK and 6(OH)Mel in the skin. These analyses are consistent with the recognized antiaging properties of melatonin [46], supported further by the functional data presented in Section 3.4. In addition, the GSEA also suggests an antiviral activity of melatonin and its metabolites in keratinocytes with upregulation of innate immunity. This is consistent with IPA, indicating a role of RIG1-like receptors in the antiviral innate immunity pathway for 6(OH)Mel as well as the role of PKR in the interferon induction and antiviral responses. The role of RIG1-like receptors in antiviral innate immunity and the role of PKR in interferon induction and antiviral response signaling were also seen after AFMK treatment. The same set of pathways was found in IPA for melatonin. These data are consistent with previous reports on antiviral properties of melatonin [38].

AFMK significantly downregulated IL-6 signaling pathway activation (Z-score = −2.121) in IPA. 6(OH)Mel also downregulated *IL6* gene expression (fold change = −7.56), decreased the expression of IL6 signal transducer (*Gp130*) (fold change = −1.16) and decreased the expression of *IL6R* (fold change = −1.11). Similarly, AFMK decreased *IL6* (fold change = −1.54), *IL6ST* (fold change = −1.06) and *IL6R* (fold change = −1.16) gene expressions. Melatonin also decreased expression of both *IL6R* (fold change = −1.29) and *IL6ST* (fold change = −1.28) genes. In GSEA, melatonin and its derivatives upregulated IL-10 signaling, which indicates their ability to suppress inflammation and immune responses (Table 4).

Melatonin and its metabolites (AFMK and 6(OH)Mel) protect primary keratinocytes against apoptosis by stimulation of expression of nuclear factor interleukin 3 (*NFIL3*) (RNA*seq* data) (Figure 1), indicating protection from programmed cell death (fold change = 1.28, = 1.11 for melatonin and AFMK, respectively) and by inhibition of apoptotic activator *BCL2L11* (fold change = −1.35 and −1.08, for AFMK and 6(OH)Mel, respectively). IPA has shown that AFMK decreased signaling of calcium-induced T lymphocyte apoptosis (Z score = −1.67) and apoptosis signaling (Z score = −0.82). These analyses are consistent with anti-apoptotic and prodifferentiation effects of melatonin and/or its metabolites on human keratinocytes [7,10,13,17,23,28,44,47].

We also found an influence of melatonin on an expression of genes involved in the regulation of molecular clock in the skin through GSEA. We noticed an overrepresentation of circadian clock genes set. Differential expression analysis of RNAseq data indicated changes in expression of particular genes associated with circadian clock under influence of melatonin. There were upregulations of the *NPAS2* (fold change = 2.3) and *PPAR*γ (fold change = 2.7) genes as well as of the *HDAC4* gene (fold change = 2.1) (Figure 1A). The activation of PPARγ leads to strong anti-inflammatory effects [48]. Other genes affected by melatonin included *ARNTL* (fold change = −1.1), *PER1* (fold change *=* 1.05), *PER3* (fold change = 1.38), *CRY1* (fold change = 1.01) and *CRY2* (fold change = 1.48), *CLOCK* (fold change = −1.01), *SIRT1* and *RORA* (fold change = 1.52). Genes typical of the circadian network in the liver that have been stimulated by melatonin in keratinocytes include *AURKB* (fold change = −1.03), *BIRC5* and *CENPA* (fold change = −1.01), *CHEK1*, *CKAP5* and *CSNK2A1* (fold change = −1.02), *HLAB*, *INCENP*, *KIF2C* and *MAD2L1* (fold change = −1.3), *MCM3* and *MCM4* (fold change = −1.1), *MCM6* (fold change = −1.2), *RFC4* (fold change = −1.2), *SFN*, *SPC24*, *SRC* and *UBD* (fold change = −1.11), and *ZWINT* (fold change = −1.1). Therefore, we suggest that melatonin can also affect the circadian molecular clock in keratinocytes in a similar manner as in other organs [49].

The epidermal lipids produced in keratinocytes play an essential role in the skin’s barrier formation and function [50]. These lipids constitute a barrier against the loss of water and electrolytes and a barrier against microbial and viral invasion. However, elevated sebum excretion with lipid content is a key factor involved in the acne pathology [50,51]. The GSEA also suggested hypolipidemic and antiatherogenic potential for melatonin and its metabolites (Table 5).

Interestingly, melatonin and 6(OH)Mel increased gamma-aminobutyric acid (GABA) receptor signaling in canonical IPA (Figure 2). The GABA receptors respond to GABA, the main inhibitory neurotransmitter in the mature vertebrate central nervous system (CNS). Therefore, we speculate that melatonin, AFMK and 6(OH)Mel might act similarly to GABA receptor agonists, which are the class of drugs that typically possesses sedative action with anticonvulsant, anxiolytic and muscle relaxant activities [52].

Finally, the analysis indicates the anticancer potential of melatonin, AFMK and 6(OH)Mel by inhibition of pathways involved in process of carcinogenesis or by activation of anticancer immune signaling (Table 6). Antitumor properties of the tested compound were confirmed by an additional pathway analysis of differential expressed genes using the reactome platform. This showed negative regulation of Notch 4 signaling (with entities ratio = 0.008, *p* value = 0.04, FDR = 0.451; reactions ratio = 4.92). These analyses indicate complex action of melatonin and its metabolites against multiple types of tumors and are consistent with well-established anticancer activity of melatonin [1,8]. Interestingly, 6(OH)Mel and AFMK show inhibition of the osteoarthritis pathway in IPA with a Z-score = −0.535 and ratio = 0.102 and Z-score = −0.688 and ratio = 0.124, respectively. This analysis is consistent with the reported protective role caused by application of melatonin in experimental models of osteoarthritis [53,54,55,56].

### 3.3. Cytokine Array Analysis

Keratinocytes are the main source of cytokines in the epidermis and secrete CSF, TNFα, IL-1, IL-6, IL-3, IL-8, TGFα, TGFβ, and PDGF [57]. They can act as a complex regulatory network in the epidermis influenced by physiological or pathological signals [58,59]. The cytokines’ activity is mediated by interaction with corresponding high affinity receptors [57]. The lower level of the expression of macrophage migration inhibitory factor (MIF) was noticed in DESeq2 analysis of RNAseq data for all tested compounds (fold change: 6(OH)Mel = −1.08, melatonin = −1.06, AFMK = −1.14). Tests using the cytokine profiler array have confirmed these results (Figure 4A). We also noticed elevated interleukin 1 receptor antagonist (IL1ra) protein expression after treatment with melatonin and with 6(OH)Mel (Figure 4B).

IL1ra is a competitive inhibitor of IL1, which participates in the process of silencing of inflammatory responses [60]. This suggests that melatonin and its derivatives can attenuate the proinflammatory conditions by increasing expression of ILra protein. Serpine family E member 1 (SERPINE1)’s protein expression was significantly elevated by 6(OH)Mel only (Figure 4C). SERPINE1 is a protein required for stimulation of keratinocyte migration during cutaneous injury repair [61]. We also noticed that 6(OH)Mel can enhance the C-X-C motif chemokine ligand 1 (CXCL1) level (Figure 4D). CXCL1 is an antimicrobial protein, which also plays a role in the antiviral innate immune response [62]. These findings are consistent with the above bioinformatic analyses and well-documented immunomodulatory and protective properties of melatonin [63].

### 3.4. Results of Telomerase Assay

The RNA*seq* data and pathway analysis (Table 4) were confirmed by measurement of telomerase activity by TRAP assay in HaCaT keratinocytes treated with melatonin and its metabolites for 24 h at a concentration of 10^−4^ M (Figure 5). The use of an immortalized keratinocyte line was dictated by its phenotypic stability during passaging. The influence of the melatonin on telomerase activity has also been indicated by GO analysis of RNA*seq* data after stimulation with melatonin, which included overrepresentation of Positive-Regulation-of-Telomerase-Activity gene set enrichment with the parameter of Rank at Max equal 4064. These data and bioinformatics analyses support previous studies showing antiaging activity of melatonin in the human skin [31,43].

### 3.5. Energy Yielding Metabolism Assays

The bioinformatics analyses presented in Section 3.1 and Section 3.2 that indicated an effect on energy yielding metabolism were substantiated by qPCR analyses. Specifically, melatonin and its hydroxymetabolites stimulated sirtuin 1 (*SIRT-1*) and *PGC1α*, while inhibiting *LDHA* gene expressions in HaCaT keratinocytes (Figure 6). The latter finding has been confirmed by inhibition of lactate production in cultured cells treated with 6(OH)Mel and AFMK (Figure 6D).

Next, we performed a comprehensive metabolic analysis using the Seahorse metabolic analyzer platform. The obtained data on the oxygen consumption rate (mitochondrial respiration, OCR, Figure 6A) and extracellular acidification rate (glycolytic function, ECAR, Figure 6B) showed consistency with the RNA*seq*, bioinformatics and functional analyses presented above. We noticed decreased basal respiration and ATP production 24 h following AFMK or 6(OH)Mel compared to control or melatonin stimulated conditions (Figure 7A). Maximal respiration, proton leak and spare respiratory capacity were unaffected by melatonin or its metabolites. Both AFMK and 6(OH)Mel, but not melatonin itself, significantly decreased glycolysis and glycolytic production in HEKn cells with a low oxygen consumption rate (Figure 7B), which is consistent with inhibition of lactate production by the compounds presented in Figure 5E. In general, these studies are in line with recognized abilities of melatonin and its metabolites to affect mitochondrial functions and their intramitochondrial metabolism [2,9,14,15,16,22,27,42,64].

While previous studies on metabolic functions were predominantly performed on isolated mitochondria, the Seahorse metabolic platform utilizes the intact cells. The reported inhibition of basal respiration and ATP production and of glycolysis by melatonin metabolites in keratinocytes is consistent with reported previously inhibition of cell proliferation by melatonin and its downstream metabolites [13,21,30,44]. However, the lack of an effect of melatonin on the above metabolic parameters requires further investigation since melatonin is metabolized rapidly within mitochondria to 6(OH)Mel and AFMK and the presented cellular effect may represent an average of both melatonin effect and that of its metabolites. This issue will require additional studies that are beyond the scope of this manuscript.

## 4. Conclusions

Comprehensive analysis of RNA*sequence* data has shown significant changes in gene expression induced by melatonin and its two indolic (6(OH)Mel) and kynuric (AFMK) metabolites with an overlapping and differential pattern. This resulted in common regulation of top diseases and disorders and of molecular and cellular functions with some additional specificity for each of the tested molecules. This is common for melatonin, 6(OH)Mel and AFMF signaling pathways, with some signaling pathways being specific for each tested molecule. These may include common and separate nuclear receptors or regulatory proteins on which these molecules will act. GSEA, Reactome FIViZ and IPA indicated activation of p53 signaling and NRF2-mediated antioxidative pathways, which is consistent with our previous studies showing protective and antioxidative functions of melatonin and its metabolites. An overrepresentation of circadian clock and antiaging signaling by melatonin derivatives is consistent with antiaging properties of the melatonin. The GO showing an activation of the keratinocyte differentiation program by melatonin and the GSEA indicating antitumor activity are consistent with functional data reported previously demonstrating antiproliferative properties of melatonin and its metabolites in skin cells. Interestingly, the GSEA and IPA indicated, respectively, antilipidemic potential and antiviral responses controlled by melatonin and its metabolites. In addition, melatonin and metabolites had an effect on cellular metabolism and mitochondria functions in keratinocytes.

It is expected that the above changes will affect keratinocyte proliferation and differentiation programs to improve epidermal barrier formation, leading to increased protection against environmental insults and different pathogens. In addition, these analyses suggest possible applications of melatonin, AFMK and 6(OH)Mel in translational medicine.

## Figures and Tables

**Figure 1 antioxidants-10-00618-f001:**
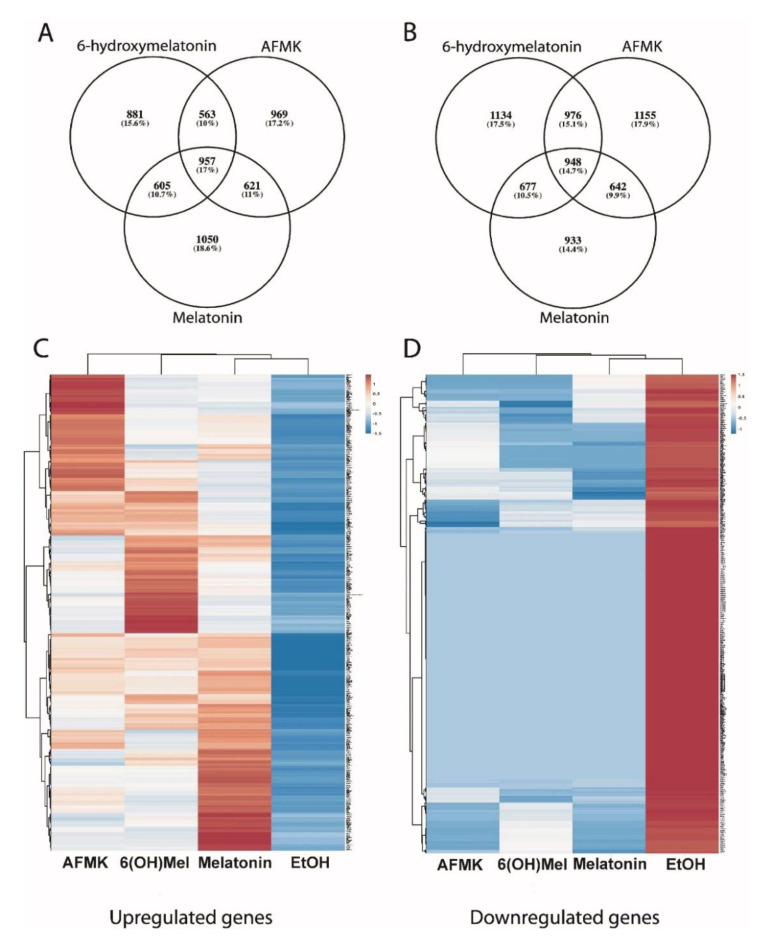
Analyses of differentially expressed genes in human keratinocytes treated with melatonin, 6-hydroxymelatonin and AFMK. Venn diagrams are in (**A**) for up- and in (**B**) for downregulated genes. Heatmaps for all upregulated genes are in (**C**) and for downregulated genes in (**D**), which are common for all tested compounds. Keratinocytes were treated with 10^−5^ M of melatonin, its metabolites or vehicle for 24 h and submitted for RNA-*sequence* analysis.

**Figure 2 antioxidants-10-00618-f002:**
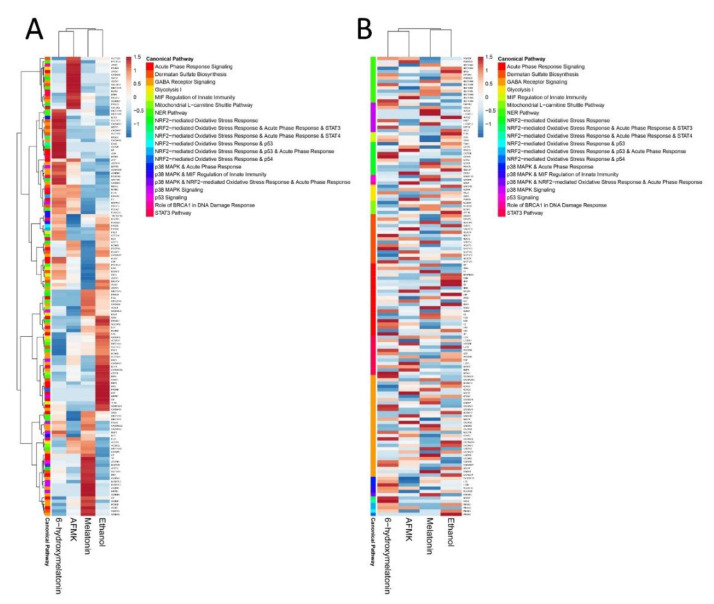
Heatmaps created from canonical pathways results of IPA with clustering (**A**) or without clustering (**B**) of the sets of the genes involved in enriched pathways.

**Figure 3 antioxidants-10-00618-f003:**
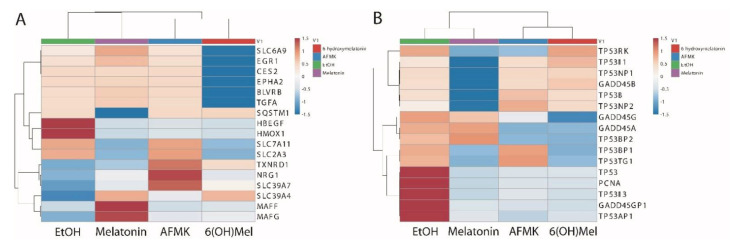
Heatmaps for NRF2 (**A**) and TP53 (**B**) signaling pathways after treatment of human epidermal keratinocytes with melatonin and its derivatives.

**Figure 4 antioxidants-10-00618-f004:**
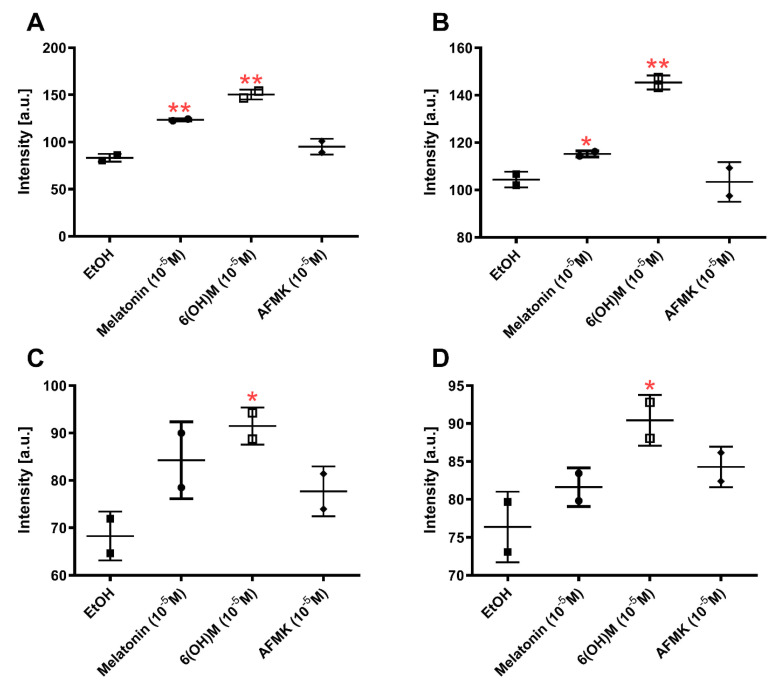
Profile of cytokine expression for melatonin, AFMK and 6-hydroxymelatonin after 24 h incubation in HEKn as results of Proteome Profiler Cytokine Array: (**A**) MIF, (**B**) IL1ra, (**C**) SERPINE1, and (**D**) CXCL1 protein expression. The plots represent all values with calculated means and significance was evaluated by Mann–Whitney U test with * *p* < 0.05 and ** *p* < 0.01.

**Figure 5 antioxidants-10-00618-f005:**
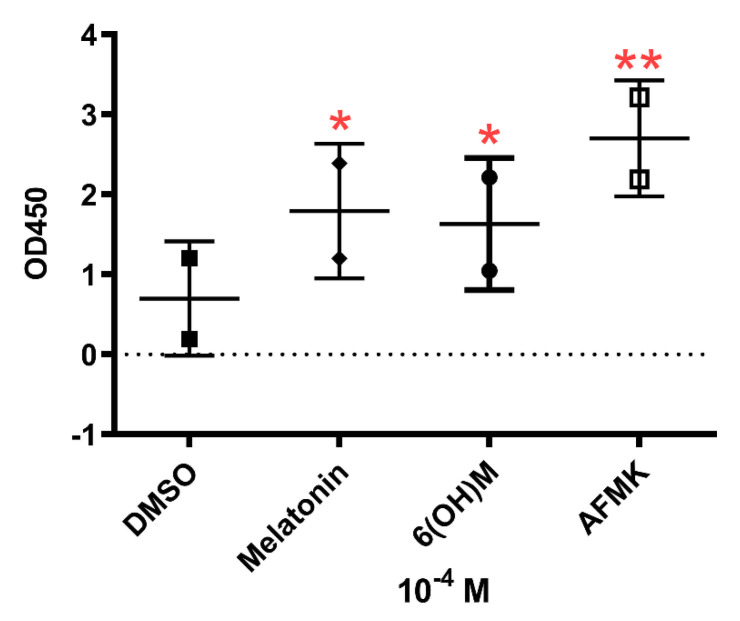
Changes in telomerase activity in epidermal HaCaT keratinocytes under influence of melatonin, 6-hydroxymelatonin or AFMK at a concentration of 10^−4^ M. The plots represent all values with calculated means and significance was evaluated by paired Student *t*-Test with * *p* < 0.05; ** *p* < 0.01.

**Figure 6 antioxidants-10-00618-f006:**
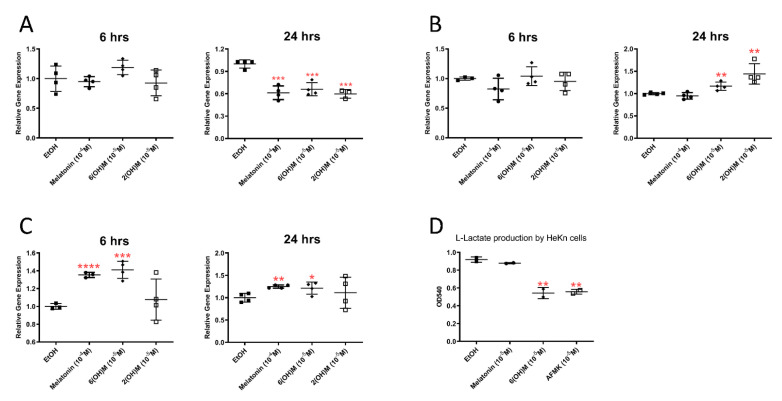
The effect of melatonin and its metabolites on the expressions of *LDHA* (**A**), *PGC-1* (**B**) and *SIRT-1* (**C**), and lactate production (**D**). Human epidermal (HaCaT) keratinocytes were cultured in presence of absence of the listed concentrations of the compounds and qPCR was performed as described in *Materials and Methods* with cyclophilin B as housekeeping gene. Each panel shows fold changes after calculated using delta-delta Ct normalized with housekeeping gene. Lactate accumulation in culture media is shown (**D**). The plots represent all values with calculated means and significance was evaluated both by Mann–Whitney U test and Student *t*-Test, which showed the same *p* values: * *p* < 0.05; ** *p* < 0.01; *** *p* < 0.001; **** *p* < 0.0001.

**Figure 7 antioxidants-10-00618-f007:**
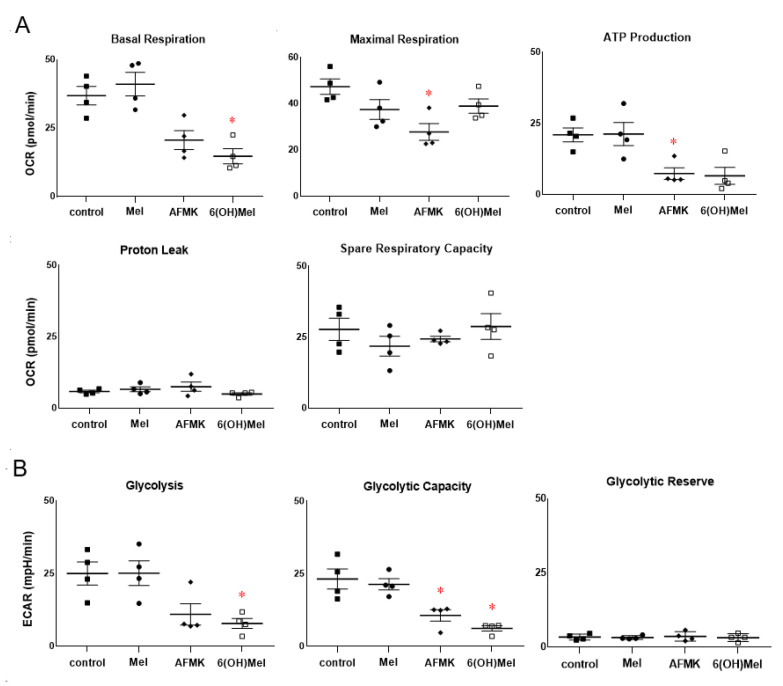
The Seahorse metabolic assay analysis performed in primary human epidermal keratinocytes treated with melatonin and its metabolites at a concentration of 10^−5^ M for 24 h. (**A**)**:** OCR and (**B**): ECAR parameters are presented along the procedure as described in *Materials and Methods*. The plots represent all values with calculated means and significance was evaluated both by Mann–Whitney U test and Student *t*-Test, which showed the same *p* values: * *p* < 0.05. Passages 2 and 3 of primary epidermal keratinocytes were used for the experiments.

**Table 1 antioxidants-10-00618-t001:** Summary of Ingenuity Pathway Analysis (IPA) for top common classes of diseases and biofunctions regulated by melatonin and its metabolites based on the data obtained from primary human epidermal keratinocytes.

Top Diseases and Biofunctions	Melatonin		AFMK		6(OH)Mel	
**Diseases and disorders**	***p* value range**	**Molecules**	***p* value range**	**Molecules**	***p* value range**	**Molecules**
Cancer	1.39 × 10^−3^ ÷ 4.30 × 10^−21^	1313	3.11 × 10^−3^ ÷ 2.06 × 10^−23^	1419	5.59 × 10^−3^ ÷ 8.62 × 10^−21^	1211
Dermatological diseases and conditions	2.99 × 10^−4^ ÷ 4.30 × 10^−21^	1073	1.60 × 10^−3^ ÷ 2.06 × 10^−23^	1127	5.22 × 10^−3^ ÷ 8.62 × 10^−21^	1065
Organismal injury and abnormalities	1.48 × 10^−3^ ÷ 4.30 × 10^−21^	1454	3.14 × 10^−3^ ÷ 2.06 × 10^−23^	1528	5.94 × 10^−3^ ÷ 8.62 × 10^−21^	1330
Reproductive system disease	1.48 × 10^−3^ ÷ 3.49 × 10^−13^	104	1.13 × 10^−10^	19	5.78 × 10^−3^ ÷ 6.37 × 10^−9^	47
Endocrine system disorders	1.47 × 10^−3^ ÷ 9.57 × 10^−13^	598	3.14 × 10^−03^ ÷ 1.56 × 10^−12^	252	6.1 × 10^−8^	194
Gastrointestinal disease	6.37 × 10^−8^	26	3.11 × 10^−3^ ÷ 1.56 × 10^−12^	404	1.6 × 10^−7^	86
Psychological disorders	1.7 × 10^−7^	18	1.35 × 10^−5^	31	5.93 × 10^−3^ ÷ 4.25 × 10^−8^	170
**Molecular and cellular functions**	***p* value range**	**Molecules**	***p* value range**	**Molecules**	***p* value range**	**Molecules**
Cell to cell signaling and interaction	1.47 × 10^−3^ ÷ 2.19 × 10^−9^	415	1.2 × 10^−5^	86	5.96 × 10^−3^ ÷ 1.71 × 10^−9^	416
Gene expression	5.80 × 10^−8^ ÷ 5.80 × 10^−8^	25	1.96 × 10^−6^	23	5.15 × 10^−5^	20
Cell signaling	1.39 × 10^−3^ ÷ 7.55 × 10^−7^	172	3.11 × 10^−3^ ÷ 2.71 × 10^−7^	191	5.96 × 10^−3^ ÷ 1.23 × 10^−8^	205
Molecular transport	1.43 × 10^−3^ ÷ 7.55 × 10^−7^	267	3.12 × 10x^−3^ ÷ 7.72 × 10^−10^	375	5.96 × 10^−3^ ÷ 3.08 × 10^−10^	349
Vitamin and mineral metabolism	7.29 × 10^−4^ ÷ 7.55 × 10^−7^	138	2.43 × 10^−3^ ÷ 2.71 × 10^−7^	150	5.96 × 10^−3^ ÷ 1.23 × 10^−8^	169
Cellular function and maintenance	1.49 × 10^−6^	69	3.12 × 10^−3^ ÷ 1.06 × 10^−9^	157	2.51 × 10^−7^	18
Cellular movement	3.95 × 10^−6^	37	2.95 × 10^−3^ ÷ 4.56 × 10^−7^	208	7.3 × 10^−6^	62
Cell morphology	9.84 × 10^−4^	7	5.36 × 10^−5^	15	5.34 × 10^−3^ ÷ 2.51 × 10^−7^	97

**Table 2 antioxidants-10-00618-t002:** Gene Set Enrichment Analysis (GSEA) results for oxidative stress and DNA damage response gene sets affected by melatonin, AFMK and 6-hydroxymelatonin (6(OH)Mel) in human epidermal keratinocytes; NES—Normalized Enriched Score; (×)—the effect is absent.

Reactome Pathway	GSEA for Melatonin		GSEA for AFMK		GSEA for 6(OH)Mel	
	NES	Direction	NES	Direction	NES	Direction
Detoxification of reactive oxygen species	1.09	↑	1.10	↑	1.10	↑
DNA double-strand break response	1.03	↑	1.04	↑	1.03	↑
Base excision repair	0.99	↑	×	×	0.99	↑
p53-dependent G1/S DNA damage checkpoint	1.11	↑	1.09	↑	×	×
DNA double-strand break repair	0.99	↑	1.00	↑	0.99	↑
p53-independent G1/S DNA damage checkpoint	1.10	↑	1.12	↑	1.11	↑
TP53 regulates transcription of DNA repair genes	1.01	↑	1.002	↑	0.99	↑
Regulation of TP53 activity through methylation	1.07	↑	1.10	↑	1.08	↑
Regulation of TP53 activity through phosphorylation	1.00	↑	×	×	×	×
Recognition of DNA damage by PCNA-containing replication complex	1.05	↑	×	×	1.06	↑
Dual incision in GG-NER	1.05	↑	×	×	×	×
Gap-filling DNA repair synthesis and ligation in GG-NER	1.05	↑	×	×	1.05	↑
PCNA-dependent long patch base excision repair	1.03	↑	×	×	×	×
DNA damage recognition in GG-NER	×	×	1.08	↑	1.07	↑
Gap-filling DNA repair synthesis and ligation in TC-NER	×	×	1.04	↑	1.04	↑
G2/M DNA damage checkpoints	×	×	1.02	↑	×	×
G1/S DNA damage checkpoints	×	×	1.09	↑	1.10	↑
Stabilization of p53	×	×	1.12	↑	×	×
DNA damage bypass	×	×	1.05	↑	×	×
Nucleotide excision repair	×	×	×	×	1.04	↑

**Table 3 antioxidants-10-00618-t003:** GSEA for mitochondrial metabolism/signaling in primary human epidermal keratinocytes treated with melatonin, 6(OH)Mel or AFMK; NES—Normalized Enriched Score; FDR—False Discovery Rate; (×)—the effect is absent.

Influence of Melatonin Metabolites on Mitochondrial Metabolism									
Reactome Pathway	GSEA for Melatonin			GSEA for 6(OH)Mel			GSEA for AFMK		
	NES	FDR	Direction	NES	FDR	Direction	NES	FDR	Direction
Citric acid cycle	1.06	0.50	↑	1.07	0.49	↑	1.06	0.52	↑
Glucose metabolism	1.06	0.50	↑	1.05	0.51	↑	1.05	0.51	↑
Mitochondrial translation	1.07	0.50	↑	1.07	0.49	↑	1.07	0.50	↑
Cristae formation	1.07	0.50	↑	1.07	0.49	↑	1.09	0.54	↑
Pyruvate metabolism and citric acid cycle	1.05	0.50	↑	1.05	0.51	↑	1.05	0.51	↑
Mitochondrial translation elongation	1.08	0.51	↑	1.09	0.57	↑	1.09	0.53	↑
Pyruvate metabolism	1.05	0.52	↑	1.03	0.53	↑	1.06	0.50	↑
Mitochondrial protein import	1.04	0.53	↑	1.05	0.51	↑	1.03	0.54	↑
Mitochondrial biogenesis	1.04	0.53	↑	1.05	0.51	↑	1.03	0.54	↑
Mitochondrial translation initiation	1.09	0.54	↑	1.07	0.48	↑	1.08	0.51	↑
Mitochondrial fatty acid β-oxidation	1.03	0.56	↑	×	×	×	×	×	×
The citric acid (TCA) cycle and respiratory electron transport	1.09	0.57	↑	1.09	0.56	↑	1.08	0.52	↑
Gluconeogenesis	1.12	0.85	↑	1.13	0.77	↑	1.12	0.86	↑
Glycogen metabolism	1.11	0.76	↑	1.11	0.72	↑	1.11	0.79	↑
Mitochondrial tRNA aminoacylation	1.00	0.61	↑	1.00	0.59	↑	1.00	0.60	↑
Respiratory electron transport	1.10	×	×	1.10	0.63	↑	1.09	0.57	↑
Respiratory electron transport, ATP synthesis by chemiosmotic coupling, and heat production by uncoupling proteins	*×*	×	×	1.10	0.68	↑	1.11	0.78	↑
Glycogen synthesis	1.10	0.67	↑	*×*	*×*	*×*	1.09	0.57	↑

**Table 4 antioxidants-10-00618-t004:** Therapeutic implications for melatonin, AFMK and 6(OH)Mel based on GSEA. NES—normalized enriched score; FDR—false discovery rate; (×)—the effect is absent.

Therapeutic Effects of Melatonin and Its Metabolites									
Reactome Pathway	GSEA for Melatonin			GSEA for 6(OH)Mel			GSEA for AFMK		
	NES	FDR	Direction	NES	FDR	Direction	NES	FDR	Direction
Antiviral mechanism by IFN-stimulated genes	*×*	*×*	*×*	1.03	0.57	↑	1.03	0.55	↑
ISG15 antiviral mechanism	1.03	0.55	↑	*×*	*×*	*×*	1.03	0.55	↑
Defensins	0.94	0.89	↑	0.94	0.90	↑	0.94	0.91	↑
Beta defensins	0.93	0.93	↑	0.92	0.95	↑	0.91	0.96	↑
Tat-mediated HIV elongation arrest and recovery	1.06	0.50	↑	1.05	0.51	↑	1.07	0.49	↑
HIV elongation arrest and recovery	1.07	0.51	↑	1.06	0.50	↑	1.08	0.49	↑
Abortive elongation of HIV-1 transcript in the absence of Tat	1.06	0.49	↑	1.06	0.50	↑	*×*	*×*	*×*
Pausing and recovery of HIV elongation	1.05	0.49	↑	1.06	0.51	↑	*×*	*×*	*×*
Telomere maintenance	0.98	0.69	↑	0.99	0.66	↑	0.99	0.64	↑
Packaging of telomere ends	1.01	0.60	↑	1.02	0.58	↑	1.03	0.55	↑
Extension of telomeres	0.96	0.83	↑	0.97	0.75	↑	0.97	0.76	↑
Telomere C-strand (Lagging Strand) synthesis	0.97	0.74	↑	0.96	0.83	↑	0.95	0.86	↑
Interleukin-10 signaling	1.06	0.49	↑	1.06	0.50	↑	1.07	0.49	↑

**Table 5 antioxidants-10-00618-t005:** Implications for cardiovascular system based on the GSEA of gene expression pattern affected by melatonin, AFMK and 6(OH)Mel in primary human epidermal keratinocytes. NES—normalized enriched score; (×)—the effect is absent.

Reactome Pathway	GSEA for Melatonin		GSEA for 6(OH)Mel		GSEA for AFMK	
	NES	Direction	NES	Direction	NES	Direction
Plasma lipoprotein assembly, remodeling, and clearance	1.08	↑	1.08	↑	1.08	↑
ABC-family proteins-mediated transport	1.08	↑	1.08	↑	*×*	*×*
Triglyceride catabolism	1.04	↑	*×*	*×*	*×*	*×*
LDL clearance	1.09	↑	1.09	↑	1.08	↑
PPARα activated gene expression	1.01	↑	*×*	*×*	1.00	↑
Assembly of active LPL and LIPC lipase complexes	1.00	↑	*×*	*×*	1.01	↑
Platelet homeostasis	1.00	↑	1.00	↑	*×*	*×*
Plasma lipoprotein clearance	*×*	*×*	*×*	*×*	1.07	↑
Prostacyclin signaling through prostacyclin receptor	*×*	*×*	1.10	↑	1.11	↑
Platelet calcium homeostasis	*×*	*×*	*×*	*×*	0.98	↑
Regulation of lipid metabolism by PPARα	*×*	*×*	*×*	*×*	0.97	↑
Nitric oxide stimulated guanylate cyclase	*×*	*×*	*×*	*×*	0.86	↑
Vasopressin regulates renal water homeostasis via aquaporins	*×*	*×*	1.04	↑	*×*	*×*
Regulation of insulin secretion	*×*	*×*	1.04	↑	*×*	*×*
Triglyceride catabolism	*×*	*×*	1.03	↑	*×*	*×*
ABC transporters in lipid homeostasis	*×*	*×*	0.89	↑	*×*	*×*

**Table 6 antioxidants-10-00618-t006:** Indications for protective effects of melatonin, AFMK and 6(OH)Mel against common human tumors based on Canonical Ingenuity Pathway Analysis (IPA) of data obtained from primary human epidermal keratinocytes. (×)—the effect is absent.

IPAs	IPA for Melatonin		IPA for AFMK		IPA for 6(OH)Mel	
	Z-Score	Ratio	Z-Score	Ratio	Z-Score	Ratio
PD-1, PD-L1 cancer immunotherapy pathway	0.33	0.11	1.51	0.15	*×*	*×*
Renal cell carcinoma signaling pathway	−2	0.06	−1.34	0.07	*×*	*×*
Basal cell carcinoma signaling	*×*	0.08	*×*	*×*	−2.24	0.14
Glioma signaling	−1.63	0.06	−0.70	0.08	*×*	*×*
Acute myeloid leukemia signaling	−1.34	0.08	−1.41	0.11	−1.34	0.08
SPINK1 general cancer pathway	*×*	*×*	−0.70	0.15	*×*	*×*
Estrogen-dependent breast cancer signaling	*×*	*×*	−1	0.12	*×*	*×*
SPINK1 pancreatic cancer pathway	*×*	*×*	−0.44	0.12	*×*	*×*
Small cell lung cancer signaling	*×*	*×*	−1	0.07	*×*	*×*
Endometrial cancer signaling	*×*	*×*	−1	0.08	*×*	*×*
Non-small cell Lung cancer signaling	*×*	*×*	−1.34	0.09	*×*	*×*
Pancreatic adenocarcinoma signaling	*×*	*×*	−1.63	0.05	*×*	*×*
Glioma Invasiveness signaling	*×*	*×*	−1.13	0.12	*×*	*×*

## Data Availability

All raw RNAseq data were deposited in NCBI GEO (GSE147588).

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
