# Peer review of "Differential and Overlapping Effects of Melatonin and Its Metabolites on Keratinocyte Function: Bioinformatics and Metabolic Analyses"

_antioxidants, 2021, doi:10.3390/antiox10040618_

Round 1

Reviewer 1 Report

Congratulations to the authors for a very thorough analysis of the melatonin effects and its metabolites on keratinocyte function.

The structure of the manuscript is correct, the content is understandable to the reader. The authors draw the right conclusions from the work on research. I have no comments.

Author Response

Congratulations to the authors for a very thorough analysis of the melatonin effects and its metabolites on keratinocyte function.

The structure of the manuscript is correct, the content is understandable to the reader. The authors draw the right conclusions from the work on research. I have no comments.

 Authors’ response: The authors would like to thank the reviewer for their positive comments regarding our study.

Reviewer 2 Report

In the present work Stefan et al. provide a comprehensive bioinformatics analysis, using RNAseq data, on the effects of melatonin and melatonin metabolites, AFMK 6(OH)Mel on cultured human epidermal keratinocytes. In silico results, related with specific function attributed to melatonin and its metabolites were further corroborated using experimental analyses in primary epidermal keratinocytes or in an immortalized keratinocyte cell line (HaCaT). Overall the result further validate the numerous functions attributed to melatonin and provide clues on the genes and gene pathways acting on such functions. Yet, the poor resolution of images makes then hard or almost impossible to read. For instance Figure 2 is unreadable.

Also in Figure 3D, it seems that you obtained a large set of genes with the same expression profile in all test conditions (uniform blue colour), it this real?

Line 59: The authors should be clearer regarding melatonin production sites: despite being synthesized in peripheral organs, such as the skin it is mostly produced in the pineal gland.

For the experimental analysis, the concentrations used are rather high, how do they compare with previous reports?

Why did the authors use an immortalized cell line for RT-qPCR (HaCaT) while RNA-seq was done in primary cells? Also, Lactate Assay kit done in HEKn but LDHA expression on HaCaT.

Remove embedded links from lines 371 to 376

Supplementary Figure 2. Move to main figure?

Line 434: Figure 5D instead of Supplementary Figure 5D?

Could you provide the primer pair efficiencies for RT-qPCR reactions?

Author Response

Reviewer #2:

1. In the present work Stefan et al. provide a comprehensive bioinformatics analysis, using RNAseq data, on the effects of melatonin and melatonin metabolites, AFMK 6(OH)Mel on cultured human epidermal keratinocytes. In silico results, related with specific function attributed to melatonin and its metabolites were further corroborated using experimental analyses in primary epidermal keratinocytes or in an immortalized keratinocyte cell line (HaCaT). Overall the result further validate the numerous functions attributed to melatonin and provide clues on the genes and gene pathways acting on such functions. Yet, the poor resolution of images makes then hard or almost impossible to read. For instance Figure 2 is unreadable.

Authors’ response: The authors greatly appreciate the reviewer’s comments and critique. The figure 2 has been amended with a better quality presentation.

2. Also in Figure 3D, it seems that you obtained a large set of genes with the same expression profile in all test conditions (uniform blue colour), it this real?

 Authors’ response: The authors appreciate the reviewer critique. The real values were transformed to color scale that show range of changes. The actual differences fall within the range of color scale.

3. Line 59: The authors should be clearer regarding melatonin production sites: despite being synthesized in peripheral organs, such as the skin it is mostly produced in the pineal gland.

 Authors’ response: This has now been corrected on page 2: “Melatonin is mainly produced in the pineal gland but can also be synthesized in the peripheral organs such as in the epidermis of the skin [13,17–22].”

4. For the experimental analysis, the concentrations used are rather high, how do they compare with previous reports?

Authors’ response: We agree. The concentrations of the ligand we used are high. However, melatonin is produced and metabolized in the epidermis. Most importantly, the reported results are consistent with our previous work, which is cited in this paper and other investigative teams working on skin cells that are also referred in the introduction and discussion. This has been further clarified on page 3: “We have used similar doses of ligands in our previous studies demonstrating the paracrine mechanism of action secondary to production and metabolism of melatonin in the epidermis [12,13,29,30].

5. Why did the authors use an immortalized cell line for RT-qPCR (HaCaT) while RNA-seq was done in primary cells? Also, Lactate Assay kit done in HEKn but LDHA expression on HaCaT.

Authors’ response: The RNA-seq was done on primary keratinocytes from different patients. The phenotype and properties of keratinocytes can differ depending on the donor and passage number. Immortalized HaCaT epidermal keratinocytes show phenotypic consistency and therefore have been used for quantitative analyses. To clarify this point following sentence was added on page 15: “The use of immortalized keratinocyte line was dictated by its phenotypic stability during passaging.

6. Remove embedded links from lines 371 to 376

Authors’ response: This has now been done. Thank you for pointing this out.

7. Supplementary Figure 2. Move to main figure?

Authors’ response: The supplementary Figure 2 has now been moved to the main text. We thank the reviewer for this request.

8. Line 434: Figure 5D instead of Supplementary Figure 5D?

Authors’ response: This has been corrected. Thank you for pointing this out.

9. Could you provide the primer pair efficiencies for RT-qPCR reactions?

Authors’ response: Thank you very much for this valuable comment. Enclosed sequences of primers were used followed by previous reports:

[LDHA https://link.springer.com/article/10.1186/ar4011; PGC1  https://www.sciencedirect.com/science/article/pii/S1357272512002749; SIRT1  https://pubmed.ncbi.nlm.nih.gov/19934257/;

cyclophilin B https://www.sciencedirect.com/science/article/pii/S2352396419304979] where their efficiencies reached 100% along the respective slope and Ct values. In addition, we analyzed pairs of primers in silico using UCSC In-Silico PCR software (Santa Cruz, CA, USA) for confirmation of the products as well as using OligoAnalyzer software (Coralville, IA, USA) where query coverage equaled 100%. Finally, the efficiencies of primers were determined by the supplier, i.e. Eurofins Genomics (Ebersberg, Germany). 

Reviewer 3 Report

The authors analyze the effects of melatonin and its metabolites on human keratinocytes. They performed an extensive analysis of RNAseq to determine which genes were up- and down-regulated and functional analysis to corroborate these results. They show that melatonin and its metabolites have protective effects by upregulating NRF2 and p53 signaling pathways, reducing inflammation, and improving telomerase activity, among others. Some aspects need to be clarified to improve the quality of the manuscript:

  1. I think that a proliferation assay is needed to better assess the impact of treatments.
  2. In figure 3, please give the results putting the treatments in the same order in heatmaps A and B. Right now it is difficult to compare the effects of treatments.
  3. The authors show an increase in the expression of SIRT1 and PGC1, as well as a decrease in LDH. This should lead to increased mitochondrial biogenesis and function. However, in the OCR experiment, the basal respiration and ATP production are greatly decreased, with no changes in proton leak. Please explain these contradictory results.
  4. According to the authors, IL-6 signaling pathway is downregulated and IL-10 is upregulated with treatments. However, in the cytokine array analysis they only give the results for MIF, IL1ra, SERPINE1, and CXCL1. Could the authors also show the results for IL-6 and IL-10?
  5. The authors use two different types of cells and they change the concentration of treatments from 10-5 M to 10-4 M. Please justify this change.
  6. Please remove the links to Wikipedia on page 14, lines 371, 373, and 376.
  7. On page 16, line 434 the authors refer to Supplementary Figure 5D. However, there are only 2 Supplementary figures.

Minor points:

  • The heatmaps in Figure 2 are illegible. Please improve the image quality.
  • There are some minor typos in the text.
  • On page 2, line 68, the authors define MT1 and MT2 abbreviations. However, these abbreviations appear before in line 54.
  • Please rephrase the sentence on page 13, lines 340-341 (“Differentiation under influence…”).

Author Response

The authors analyze the effects of melatonin and its metabolites on human keratinocytes. They performed an extensive analysis of RNAseq to determine which genes were up- and down-regulated and functional analysis to corroborate these results. They show that melatonin and its metabolites have protective effects by upregulating NRF2 and p53 signaling pathways, reducing inflammation, and improving telomerase activity, among others.

Some aspects need to be clarified to improve the quality of the manuscript:

1. I think that a proliferation assay is needed to better assess the impact of treatments.

Authors’ response: We acknowledge the reviewers comment however the reason we did not include this data in this manuscript is because we have published this data in a number of previous papers hence the results are known and accepted and we cite these papers in this manuscript. To make this point more clearly to the readers, we have now added the following statement on page 18: “...with functional data reported previously demonstrating anti-proliferative properties of melatonin and its metabolites in skin cells.

2. In figure 3, please give the results putting the treatments in the same order in heatmaps A and B. Right now it is difficult to compare the effects of treatments.

 Authors’ response: Figure 3 has now been amended for easier interpretation of results. Thank you for your request.

3. The authors show an increase in the expression of SIRT1 and PGC1, as well as a decrease in LDH. This should lead to increased mitochondrial biogenesis and function. However, in the OCR experiment, the basal respiration and ATP production are greatly decreased, with no changes in proton leak. Please explain these contradictory results.

 Authors’ response: These observations are not so contradictory as the phenomenon of ‘uncoupled respiration’ has previously been reported in cardiovascular literature (Cheng et al., 2017; Adv Exp Med Biol. 2017; 982: 359–370). Here, the pertinant question is the exact mechanism by which proton leak occurs and it has been proposed that only the large proton leak induces opening of MPTP which is detrimental to mitochondria and adversely impacts cell viability. This mechanisms remains to be tested in skin cells however it may be one possibility of the uncoupling between basal respiration and proton leak (well described by Cheng et al., 2017; Adv Exp Med Biol. 2017; 982: 359–370).

4. According to the authors, IL-6 signaling pathway is downregulated and IL-10 is upregulated with treatments. However, in the cytokine array analysis they only give the results for MIF, IL1ra, SERPINE1, and CXCL1. Could the authors also show the results for IL-6 and IL-10?

Authors’ response: For measurement of cytokines expression, the Proteome Profiler Human Cytokine Array Kit (R&D systems) was used. The array contained antibodies against IL-6 and IL-10. However, the expression of IL-6 and IL-10 in the cell culture media was below detectability level using this kit.

5. The authors use two different types of cells and they change the concentration of treatments from 10-5M to 10-4 Please justify this change.

Authors’ response: The concentrations used were based on the previous studies that have established that 10-5 and 10-4 M concentrations of melatonin or it metabolites were optimal to exert their desirable phenotypic effects in keratinocytes. Furthermore, the phenotype and properties of primary keratinocytes can differ depending on the donor, their skin color and passage number of the cells, while immortalized HaCaT epidermal keratinocytes show phenotypic consistency and therefore have been used for detailed quantitative analyses. These results are consistent with our and other published results on skin cells, which are listed in this paper. To make these points clear we introduced following clarifications:

Page 3: “We have used similar doses of ligands in our previous studies demonstrating the paracrine mechanism of action secondary to production and metabolism of melatonin in the epidermis [12,13,29,30].

Page 18: “...with functional data reported previously demonstrating anti-proliferative properties of melatonin and its metabolites in skin cell.

6. Please remove the links to Wikipedia on page 14, lines 371, 373, and 376.

 Authors’ response: This has been done.

7. On page 16, line 434 the authors refer to Supplementary Figure 5D. However, there are only 2 Supplementary figures.

Authors’ response: We thank the reviewer for pointing this out. This error has now been corrected. It was figure 5D.

 Minor points:

  • The heatmaps in Figure 2 are illegible. Please improve the image quality.
  • There are some minor typos in the text.
  • On page 2, line 68, the authors define MT1 and MT2 abbreviations. However, these abbreviations appear before in line 54.
  • Please rephrase the sentence on page 13, lines 340-341 (“Differentiation under influence…”).

Authors’ response: All “minor points” have been addressed and corrected. Thank you for pointing them out.